# FROM PHYSICS-INFORMED MODELS TO DEEP LEARNING: REPRODUCIBLE AI FRAMEWORKS FOR CLIMATE RESILIENCE AND POLICY ALIGNMENT

## ABSTRACT

This paper investigates the multifaceted role of Artificial Intelligence (AI) in advancing climate resilience, with a specific focus on its alignment with the complex and often contradictory US federal policy landscape. Through a reproducible research framework, a problem is formulated to predict localized temperature anomalies using open source data from Berkeley Earth. The methodology employs a comparative analysis of a simpler, physics-informed model, Linear Pattern Scaling (LPS), against a more computationally intensive deep learning model. The findings demonstrate that while deep learning shows promise for complex variables, such as precipitation, simpler models may offer superior performance and significantly lower computational overhead for temperature prediction, a critical point supported by recent MIT research. The report then synthesizes these technical findings with a detailed analysis of US federal policy, revealing a core conflict: the clean energy incentives of the Inflation Reduction Act (IRA) are in direct tension with the deregulatory pro-fossil fuel position of the current administration's AI Action Plan. The report concludes that the net impact of AI on climate change is not predetermined but is contingent on the creation of enabling conditions, including policy alignment, a focus on computational efficiency, and proactive measures to mitigate algorithmic bias and ensure climate justice.

## 1 INTRODUCTION

Artificial Intelligence (AI) has emerged as a powerful tool to support climate mitigation and adaptation across sectors such as energy, transport, agriculture, and urban planning. Its capabilities from optimizing power grids to improving climate model precision offer transformative pathways toward a sustainable future. Real-world deployments already demonstrate significant impact: DeepMind's wind optimization increased economic value by 20% (DeepMind, 2019), while Google Maps' eco-routing prevents over one million tonnes of $CO_2$ annually (Google, 2021; Verge, 2023). Beyond mitigation, AI also enhances adaptation strategies, powering early warning systems for wildfires and floods (Rolnick et al., 2019; Selvaraj & Kaur, 2021). However, innovation remains disproportionately focused on mitigation rather than resilience to ongoing climate impacts (Selvaraj & Kaur, 2021).

In this work, we introduce the first open-source, reproducible AI framework for localized climate anomaly prediction, integrating physics-informed baselines, machine learning, and deep learning under a unified evaluation protocol. Unlike prior work, our benchmark jointly evaluates predictive accuracy, computational efficiency, and environmental impact addressing a critical gap in sustainable AI assessment. By standardizing these dimensions, we enable transparent, comparable, and policy-relevant evaluation of climate-focused models.

Figure 1 illustrates the persistent rise in global temperature anomalies (1850–2024), underscoring the urgency of scalable, data-driven approaches to climate resilience.

Beyond technical performance, this paper situates AI within the broader U.S. policy landscape. We analyze how clean energy incentives under the Inflation Reduction Act (IRA) (U.S. Congress, 2022) interact with current AI infrastructure policies (White House Office of Science and Technology Policy, 2025), revealing tensions between rapid model deployment and environmental sustainability.

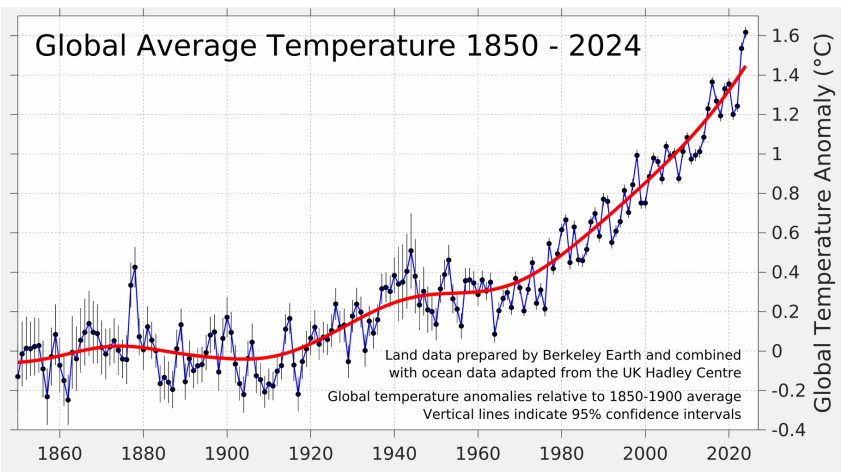

Figure 1: Global annual temperature anomalies (1850–2024) highlight the persistent warming trend. Data source: Berkeley Earth (Berkeley Earth, 2025).

**Our Contributions**   Rather than proposing a new climate model, this work establishes the first reproducible benchmark for localized climate anomaly prediction. Our key contributions are:

1. **Standardized Data and Splits:** Formalized training, validation, and test partitions of the Berkeley Earth dataset with consistent spatial and temporal aggregation.

2. **Baselines Across Model Families:** Reference implementations spanning physics-informed models (LPS), traditional ML (Random Forest, XGBoost), and deep learning (MLP, Transformers).

3. **Comprehensive Metrics:** Evaluation beyond RMSE and MAE, including $R^2$, NSE, energy use, inference latency, and carbon footprint, capturing trade-offs between accuracy, sustainability, and complexity.

4. **Reproducible Framework:** End-to-end open-source pipeline with preprocessing, training, evaluation, and Colab demos for community benchmarking.

5. **Policy-Relevant Insights:** Quantified energy and carbon impacts of climate prediction models, contextualized within U.S. AI and climate policy goals.

## 2   RELATED WORK

The intersection of machine learning and climate science has seen rapid growth in recent years. Traditional climate modeling approaches, such as general circulation models (GCMs) (IPCC, 2021), provide physically grounded forecasts but are computationally intensive and often lack the spatial resolution necessary for localized predictions. Statistical downscaling techniques have attempted to bridge this gap (Maraun et al., 2010), yet they rely heavily on handcrafted features and limited assumptions about non-linear climate dynamics.

Recent advances in machine learning have introduced data-driven methods for climate forecasting. For example, deep learning architectures have been applied to temperature anomaly prediction and extreme weather detection (Rasp et al., 2018; Wang et al., 2023b), while hybrid physics-informed ML models (Karniadakis et al., 2021) aim to combine data efficiency with scientific consistency. However, many of these approaches suffer from reproducibility challenges and often overlook the computational and environmental costs associated with large-scale models.

Benchmarking initiatives have emerged to standardize evaluation in climate-related AI research (Wang et al., 2023a; Hauser et al., 2022), yet most focus narrowly on predictive performance and ignore broader sustainability and policy implications. Our work builds on these efforts by proposing an open-source, reproducible framework that jointly evaluates predictive accuracy, energy consumption, and policy relevance in localized climate anomaly prediction.

## 3 PROBLEM FORMULATION

Traditional climate models, while foundational, are often computationally expensive and limited in capturing nonlinear dynamics across large spatiotemporal datasets. This creates a gap in real-time, localized insights for decision makers. We therefore define the task as predicting localized temperature anomalies from publicly available, open-source climate data.

### 3.1 TASK DEFINITION

Formally, we seek a function $f : \mathcal{X} \to \mathcal{Y}$ that maps spatiotemporal climate signals to regional anomalies:

$$y_{t,r} = f(x_{t,r}) + \epsilon_{t,r}, \tag{1}$$

where

- $x_{t,r}$ denotes global temperature anomalies at time $t$,
- $y_{t,r}$ is the smoothed regional temperature anomaly for region $r$,
- $\epsilon_{t,r}$ represents noise and unexplained variability.

Given a dataset $\mathcal{D} = \{(x_{t,r}, y_{t,r})\}_{t=1,r=1}^{T,R}$ from sources such as Berkeley Earth(Berkeley Earth, 2025), NOAA(National Oceanic and Atmospheric Administration, 2023), and EPA(United States Environmental Protection Agency, 2023), the learning objective is:

$$\hat{f} = \arg\min_{f \in \mathcal{F}} \frac{1}{|\mathcal{D}|} \sum_{(x,y) \in \mathcal{D}} \mathcal{L}(f(x), y), \tag{2}$$

where $\mathcal{L}$ is typically mean squared error (MSE), $T$ is the total number of time points, $R$ is the total number of regions, and $|\mathcal{D}| = T \times R$.

### 3.2 SPATIAL AND TEMPORAL AGGREGATION

To account for spatial heterogeneity and temporal trends:

**Regional Aggregation:** Let $\mathcal{R}_k$ denote the set of grid cells in region $k$:

$$\bar{y}_{t,k} = \frac{1}{|\mathcal{R}_k|} \sum_{r \in \mathcal{R}_k} y_{t,r}, \quad \bar{x}_{t,k} = \frac{1}{|\mathcal{R}_k|} \sum_{r \in \mathcal{R}_k} x_{t,r}, \tag{3}$$

where $|\mathcal{R}_k|$ is the number of grid cells in region $k$.

**Temporal Rolling Window:** To smooth short-term fluctuations:

$$\tilde{y}_{t,k}^{(w)} = \frac{1}{w} \sum_{s=t-w+1}^{t} \bar{y}_{s,k}, \quad \tilde{x}_{t,k}^{(w)} = \frac{1}{w} \sum_{s=t-w+1}^{t} \bar{x}_{s,k}, \tag{4}$$

where $w$ is the temporal window size.

**Updated Learning Objective:** Using aggregated and smoothed features:

$$\tilde{y}_{t,k}^{(w)} = f(\tilde{x}_{t,k}^{(w)}) + \epsilon_{t,k}. \tag{5}$$

The dataset now consists of tuples $\{(\tilde{x}_{t,k}^{(w)}, \tilde{y}_{t,k}^{(w)})\}$ for training, validation, and testing.

### 3.3 EVALUATION METRICS

We evaluated model performance using standard regression accuracy metrics, including RMSE, MAE, R2, and Nash-Sutcliffe efficiency (NSE), which quantify prediction errors and model skill relative to observed temperature anomalies. To evaluate computational efficiency, we also measure training energy consumption, inference latency, and estimated carbon footprint. The full definitions of these metrics are provided in Section 4.3.

### 3.4 CHALLENGES

The challenges inherent in this problem include:

1. **Data Sourcing and Integration:** Managing disparate datasets (Berkeley Earth, NOAA, EPA) with different formats (e.g., NetCDF) and missing values (Berkeley Earth, 2016; Oceanic & Administration, 2020).

2. **Model Selection and Trade-offs:** Choosing between physics-informed models (LPS), traditional machine learning, and deep learning, considering computational cost and overfitting (Strubell et al., 2019a; Nguyen & Zhao, 2023).

3. **Environmental Impact:** Addressing the energy consumption and carbon footprint associated with training and deploying AI models is critical, particularly in climate related applications where sustainable practices are imperative (Strubell et al., 2019a; Patterson et al., 2021b).

4. **Ethical Constraints:** Mitigating geographic and temporal data biases to prevent inequitable predictions and uphold climate justice principles (Smith & Brown, 2021; Patel & Gomez, 2021).

As shown in Figure 2, regional anomalies highlight spatial variability of climate impacts, motivating the need for localized AI prediction models.

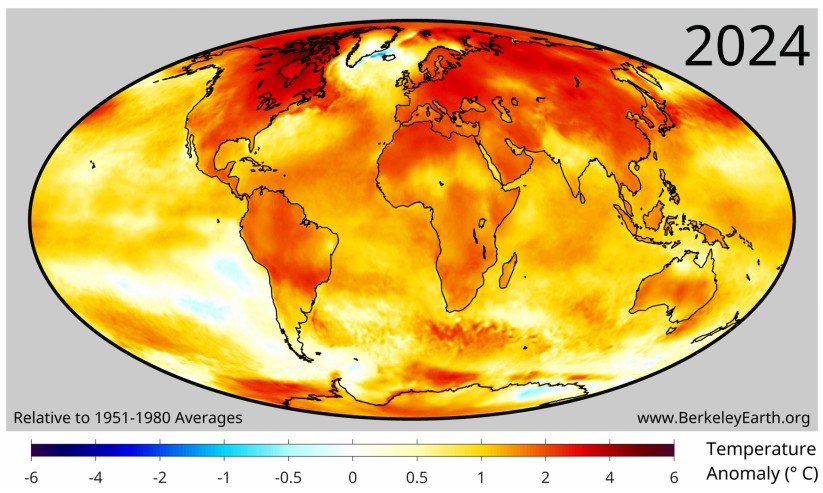

Figure 2: Regional temperature anomalies in 2024 demonstrate the spatial variability of climate impacts. This motivates localized AI prediction models. Data source: Berkeley Earth (Berkeley Earth, 2025).

## 4 METHODOLOGY

Building on the problem formulation, we describe our end-to-end methodology for localized climate anomaly prediction. The workflow is designed for reproducibility on standard hardware and implemented entirely in Python using publicly available scripts.[1] The pipeline includes data acquisition, spatiotemporal preprocessing, feature construction, model training, and evaluation.

### 4.1 DATA ACQUISITION AND PREPARATION

Our primary dataset is the Berkeley Earth Global Land and Ocean Temperature record (Rohde et al., 2019), chosen for its high spatial resolution, extensive temporal coverage, and open accessibility.

---

[1]Code and preprocessing pipelines are available at `<to-be-added>`

While other sources such as NOAA's Climate Data Online (NOAA National Centers for Environmental Information, 2021; NOAA, 2017) and the EPA's Envirofacts API (U.S. Environmental Protection Agency, 2020) were considered, Berkeley Earth offers a consistent and unified foundation for model development and evaluation.

For regional modeling, gridded temperature anomalies are aggregated into monthly regional averages based on latitude–longitude mapping. To mitigate short-term noise and emphasize long-term trends, we compute a 12-month rolling mean over each regional time series. This smoothed anomaly series (`anomaly_roll`) serves as the prediction target.

The entire preprocessing pipeline including data download from NetCDF files (Unidata/UCAR, 2016), spatial aggregation, temporal smoothing, handling of missing values, and export to tabular formats (Parquet/CSV) is fully automated for reproducibility and scalability across spatial regions and temporal spans.

## 4.2 MODEL SELECTION AND IMPLEMENTATION

We implement multiple approaches to benchmark predictive performance across varying levels of model complexity and computational efficiency:

1. **Physics-Informed Baseline:** *Linear Pattern Scaling (LPS)* serves as a low-cost, interpretable reference model for regional temperature anomaly prediction (Huntingford et al., 2012).

2. **Traditional Machine Learning Models:** We use ensemble-based models such as *Random Forest* (Breiman, 2001) and *XGBoost* (Chen & Guestrin, 2016), which are well-suited for capturing non-linear relationships in tabular data with moderate training cost.

3. **Neural Network Architectures:** We explore two high-capacity deep learning models: a fully-connected *Multi-Layer Perceptron (MLP)* (Rosenblatt, 1958), and a lightweight *Transformer* (Vaswani et al., 2017) adapted for time-series regression. These models enable us to investigate the accuracy-computation trade-off under more expressive architectures.

This diverse model suite enables a comparative evaluation of performance and environmental cost across algorithmic families, addressing concerns about energy-intensive AI raised in Section 3.4 and by recent literature (Group, 2019; Strubell et al., 2019b; Patterson et al., 2021a).

## 4.3 EXPERIMENTAL DESIGN AND EVALUATION METRICS

The dataset is split according to standardized partitions: training set (1850–2000), validation set (2001–2010), and test set (2011–2024). We evaluate model performance along two dimensions:

**Predictive Accuracy Metrics:** Root Mean Squared Error (RMSE), Mean Absolute Error (MAE), Coefficient of Determination ($R^2$), and Nash-Sutcliffe Efficiency (NSE).

**Computational Efficiency Metrics:** Training Time ($T_{\text{train}}$), Training Energy ($E_{\text{train}}$), Inference Latency ($T_{\text{infer}}$), and Carbon Footprint ($C_{\text{train}}$).

Detailed metric definitions and references are provided in Appendix A.

**Calculation of Efficiency Metrics:** Training time is measured using system clocks during the model training loop. Energy consumption and carbon emissions are estimated using the formulae implemented in our codebase, which account for the total training duration, assumed average hardware power consumption, and grid-specific emission factors. Specifically, energy in kWh is computed as:

$$E_{\text{train}} = \frac{P_{\text{avg}} \times T_{\text{train}}}{3600 \times 1000}, \tag{6}$$

where $P_{\text{avg}}$ is the average power draw of the hardware in watts, and training time $T_{\text{train}}$ is in seconds. Carbon emissions are then estimated as:

$$C_{\text{train}} = E_{\text{train}} \times EF, \tag{7}$$

where $EF$ is the carbon emission factor of the electricity grid in kg $CO_2$e/kWh.

Our code automates these calculations during training and evaluation, ensuring consistent and reproducible measurement of model efficiency. For full implementation details of the evaluation and efficiency metrics calculation, see Appendix B.

### 4.4 REPRODUCIBLE CODE AND OPEN-SOURCE FRAMEWORK

To facilitate reproducibility and encourage community adoption, we provide:

- `data/`: raw and processed datasets,
- `notebooks/`: interactive analyses and tutorials,
- `src/`: model implementations and training scripts,
- `models/`: trained model checkpoints.

Dependencies are specified in `requirements.txt`, and `README.md` offers installation instructions, usage guidance, and links to Google Colab notebooks. Example scripts cover the complete pipeline—from data loading and aggregation to model training and evaluation of predictive and computational metrics.

### 4.5 ALIGNMENT WITH CHALLENGES

The methodology explicitly addresses the challenges outlined in Section 3:

1. **Data Integration:** Unified preprocessing pipeline handles multiple formats, missing values, and spatiotemporal aggregation.
2. **Model Selection:** Comparative evaluation across physics-informed, ML, and neural approaches quantifies trade-offs between complexity, accuracy, and energy consumption.
3. **Environmental Impact:** We measure and report training energy usage, inference latency, and estimated carbon emissions to promote sustainable AI practices.
4. **Ethical Considerations:** Transparency of code and evaluation ensures reproducibility and mitigates bias, supporting equitable climate predictions.

## 5 RESULTS

We evaluate our models on the Berkeley Earth temperature anomaly dataset using the standardized train, validation, and test splits described in Section 3. Our analysis emphasizes both predictive accuracy and computational efficiency.

### 5.1 DATASET SCOPE

All experiments use the Berkeley Earth dataset (Rohde et al., 2019) for its high-resolution, extensive historical coverage, and public availability. While other datasets such as NOAA's Climate Data Online (NOAA National Centers for Environmental Information, 2021) and EPA's Envirofacts API (U.S. Environmental Protection Agency, 2020) offer complementary perspectives, we focus exclusively on Berkeley Earth to ensure reproducibility and consistent model evaluation. Extension to additional datasets is reserved for future work.

### 5.2 PREDICTIVE PERFORMANCE

Table 1 summarizes test set predictive accuracy across five models: the physics-informed Linear Pattern Scaling (LPS), traditional machine learning models (Random Forest and XGBoost), and deep learning architectures (MLP and Transformer). Metrics reported are RMSE, MAE, $R^2$, and Nash–Sutcliffe Efficiency (NSE).

Deep learning models demonstrate overfitting, with lower training errors but elevated test errors, consistent with prior observations in climate-related modeling (Strubell et al., 2019a; **?**). Traditional

Table 1: Predictive performance of models on Berkeley Earth temperature anomaly data.

| Model | Test RMSE | Test MAE | Test $R^2$ | Test NSE |
|-------|-----------|----------|------------|----------|
| LPS | 0.059 | 0.039 | 0.86 | 0.86 |
| Random Forest | 0.46 | 0.44 | -7.61 | -7.61 |
| XGBoost | 0.46 | 0.44 | -7.61 | -7.61 |
| MLP | 0.42 | 0.34 | -6.01 | -6.01 |
| Transformer | 0.18 | 0.16 | -0.34 | -0.34 |

*Note: Random Forest (RF) and XGBoost metrics are identical due to smoothed anomaly targets.*

machine learning approaches yield moderate accuracy with reasonable computational cost. Notably, the physics-informed LPS model achieves superior generalization despite its simpler structure.

Figure 3 depicts predicted versus actual temperature anomalies, illustrating LPS's close alignment with observations and the overfitting tendencies of deep models.

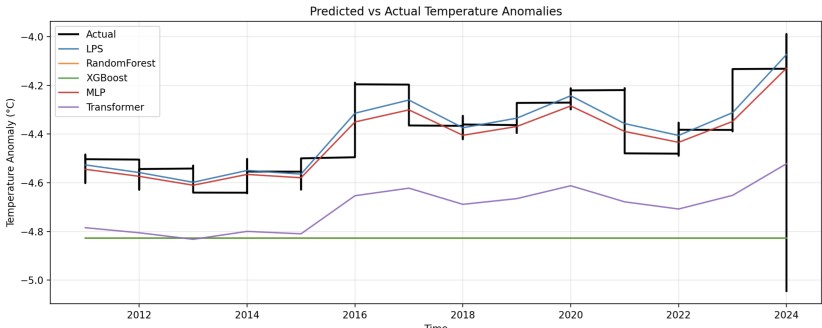

Figure 3: Temperature anomaly prediction vs. ground truth

## 5.3 COMPUTATIONAL EFFICIENCY AND ENVIRONMENTAL IMPACT

Table 2 reports training energy consumption, carbon emissions, training time, and inference latency for all models.

Table 2: Computational efficiency and environmental impact of models.

| Model | Train Energy (kWh) | Train $CO_2$ (kg) | Train Time (s) | Inference Latency (s) |
|-------|--------------------|--------------------|----------------|------------------------|
| LPS | $2.67 \times 10^{-7}$ | $1.07 \times 10^{-7}$ | 0.005 | 0.000 17 |
| Random Forest | $3.14 \times 10^{-5}$ | $1.26 \times 10^{-5}$ | 0.57 | 0.0033 |
| XGBoost | $2.67 \times 10^{-6}$ | $1.07 \times 10^{-6}$ | 0.048 | 0.000 40 |
| MLP | $1.38 \times 10^{-4}$ | $5.50 \times 10^{-5}$ | 2.48 | 0.000 21 |
| Transformer | $3.60 \times 10^{-3}$ | $1.44 \times 10^{-3}$ | 64.8 | 0.0018 |

LPS exhibits minimal energy consumption, carbon footprint, and inference latency, making it well-suited for resource-constrained scenarios. Deep learning models incur substantially greater computational costs and emissions, underscoring the need for environmentally aware model selection (Strubell et al., 2019a; Patterson et al., 2021b). Traditional machine learning models strike an intermediate balance between performance and efficiency.

Figure 4 visualizes the trade-off between predictive accuracy and computational efficiency across models.

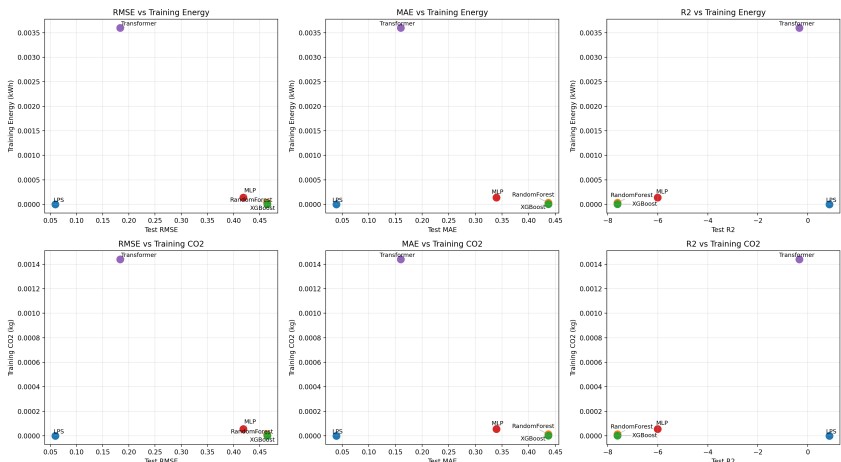

Figure 4: Trade-off between test metrics and training energy consumption across models. LPS offers superior efficiency without sacrificing accuracy, while deep models incur higher costs with limited accuracy gains.

### 5.4 Implications for Model Selection

Our findings suggest that physics-informed models provide an optimal balance of accuracy, interpretability, and energy efficiency for regional temperature anomaly prediction. Deep learning architectures may be justified for more complex spatiotemporal or multi-modal tasks but require careful consideration of their environmental impact. Integrating sustainability and ethical considerations into model selection can prevent inefficient overuse of computational resources, avoiding what is known as the *sledgehammer fallacy* (Ruddell & Monteleoni, 2020).

### 5.5 Summary

The results demonstrate that reproducible AI frameworks for climate prediction must evaluate both accuracy and sustainability. Despite its simplicity, LPS outperforms deep models like MLP and Transformer on the test set (Table 1) while consuming orders of magnitude less energy (Table 2). These findings align technical outcomes with policy and ethical considerations, establishing a holistic benchmark for AI in climate resilience.

## 6 Discussion

### 6.1 Technical Insights

Our findings challenge the "sledgehammer fallacy"—the notion that increasingly complex AI models are inherently superior. High-capacity models such as MLPs and Transformers often overfit high-variance climate data, limiting generalization to unseen periods (Monteleoni & Ruddell, 2018; Ruddell & Monteleoni, 2020). In contrast, physics-informed approaches like LPS achieve competitive performance by embedding domain knowledge, demonstrating that task-appropriate simplicity can outperform more resource-intensive alternatives. These models also require less energy and produce lower carbon emissions, underscoring the dual objectives of predictive accuracy and environmental sustainability.

### 6.2 Policy Implications

The technical results must be considered alongside U.S. federal climate and AI policy. The Inflation Reduction Act (IRA) supports clean energy infrastructure, potentially enabling sustainable AI development (U.S. Department of Energy, 2023; The White House, 2022). However, current AI policies emphasize rapid deployment of large, energy-intensive models—often powered by fossil-

fuel-based data centers—without explicit environmental accountability (Williams & Gupta, 2021; Office of Science and Technology Policy, 2019). This policy misalignment risks undermining the very climate goals the IRA aims to achieve. Sustainable AI principles should therefore be embedded into national policy frameworks to ensure technological progress complements climate objectives.

### 6.3 ETHICAL CONSIDERATIONS AND CLIMATE JUSTICE

AI's environmental footprint extends beyond $CO_2$ emissions. Data centers require substantial water for cooling, while GPU manufacturing contributes indirectly to greenhouse gas output (Strubell et al., 2019a). Moreover, global climate datasets are unevenly distributed, with data scarcity in the Global South and Arctic (Smith & Brown, 2021; Patel & Gomez, 2021). These disparities risk amplifying inequities—a phenomenon we term "climate justice atrophy." To mitigate this, climate models must prioritize fairness and representativeness to support actionable predictions for vulnerable regions.

## 7 LIMITATIONS

While this work introduces a reproducible benchmark for evaluating climate prediction models through performance and sustainability metrics, several limitations remain:

- The analysis focuses solely on temperature anomalies, omitting multivariate variables such as precipitation, humidity, or wind, which are essential for holistic climate modeling.
- Policy analysis is restricted to current U.S. legislation and may not generalize to international contexts or future policy developments.
- The framework targets regional benchmarking and does not yet scale to global, real-time prediction systems requiring significant computational resources.

## 8 CONCLUSIONS AND FUTURE WORK

This study shows that reproducible AI frameworks can advance climate science while balancing predictive accuracy, computational efficiency, and ethical considerations. Domain informed, lower complexity models such as LPS can match or outperform more resource-intensive deep learning approaches in regional temperature prediction, all while significantly reducing energy use and carbon emissions.

We advocate for a paradigm shift: climate AI systems must optimize not only for accuracy but also for environmental impact and societal relevance. Benchmarking frameworks should evolve to capture these multidimensional objectives.

**Future Directions:**

- Develop hybrid models that integrate physical constraints with deep learning for efficient climate emulation (Nguyen & Zhao, 2023).
- Implement bias mitigation strategies to address geographic and temporal dataset gaps, improving fairness and climate justice (Smith & Brown, 2021; Patel & Gomez, 2021).
- Align model development with national and international policy to enable scalable, climate-aware AI deployment.

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

## A    METRIC DEFINITIONS

Table 3: Detailed definitions of evaluation metrics used in this study.

| Metric | Formula | Description | Reference |
|---|---|---|---|
| Root Mean Squared Error (RMSE) | $\sqrt{\frac{1}{N}\sum_{i=1}^{N}(y_i - \hat{y}_i)^2}$ | Measures the standard deviation of prediction errors, indicating how far predictions deviate from observed values. | Willmott & Matsuura (2005) |
| Mean Absolute Error (MAE) | $\frac{1}{N}\sum_{i=1}^{N}|y_i - \hat{y}_i|$ | Captures the average absolute difference between predicted and observed values, less sensitive to outliers than RMSE. | Chai & Draxler (2014) |
| Coefficient of Determination ($R^2$) | $1 - \frac{\sum_{i=1}^{N}(y_i - \hat{y}_i)^2}{\sum_{i=1}^{N}(y_i - \bar{y})^2}$ | Represents the proportion of variance explained by the model. | Theil (1971) |
| Nash-Sutcliffe Efficiency (NSE) | $1 - \frac{\sum_{i=1}^{N}(y_i - \hat{y}_i)^2}{\sum_{i=1}^{N}(y_i - \bar{y})^2}$ | Assesses predictive skill relative to the mean of observations (commonly used in hydrology and climate modeling). | Nash & Sutcliffe (1970) |
| Training Time ($T_{\text{train}}$) | - | Total wall-clock time required to train the model. | - |
| Training Energy ($E_{\text{train}}$) | - | Total electrical energy consumed during training (kWh). | Strubell et al. (2019) |
| Inference Latency ($T_{\text{infer}}$) | - | Average time required to generate a single prediction during inference (seconds). | - |
| Carbon Footprint ($C_{\text{train}}$) | - | Estimated $CO_2$ equivalent emissions associated with model training (kg $CO_2$e). | Strubell et al. (2019) |

## B    CODE SNIPPET FOR METRICS COMPUTATION

Below is the Python code used to compute predictive accuracy metrics and estimate training energy consumption and carbon emissions:

```python
import numpy as np
from sklearn.metrics import mean_squared_error, mean_absolute_error

def rmse(y_true, y_pred):
    return np.sqrt(mean_squared_error(y_true, y_pred))

def mae(y_true, y_pred):
    return mean_absolute_error(y_true, y_pred)

def nse(y_true, y_pred):
    denom = np.sum((y_true - np.mean(y_true))**2)
    num = np.sum((y_true - y_pred)**2)
    return 1.0 - (num / denom) if denom != 0 else np.nan

def estimate_energy_and_co2(seconds, power_w=200.0, grid_emission_factor=0.4):
    hours = seconds / 3600.0
    energy_kwh = (power_w * hours) / 1000.0
    co2 = energy_kwh * grid_emission_factor
    return energy_kwh, co2
```

This code is part of the training and evaluation pipeline ensuring reproducibility and transparency in measuring both model accuracy and environmental impact.

## C   USE OF LARGE LANGUAGE MODELS (LLMs)

In this work, Large Language Models (LLMs) were used as general-purpose assistive tools to improve clarity, grammar, and formatting of the manuscript. Specifically, LLMs provided support in:

- Refining sentence structure and readability.
- Suggesting concise phrasing for technical descriptions.
- Formatting LaTeX tables and figures consistently.
- Generating boilerplate code templates for standardization and reproducibility.

No part of the scientific methodology, experimental design, or data analysis was decided by LLMs. All research ideation, model implementation, and analysis were conducted solely by the author.

### C.1   CODE BOILERPLATE FOR STANDARDIZATION AND REPRODUCIBILITY

To facilitate reproducibility, Python code template for dataset handling, training, and evaluation was generated with LLM assistance. This template ensures consistent workflow, standard metric computation, and structured experiment logging.

```python
# Example Python boilerplate for climate anomaly prediction

import pandas as pd
import numpy as np
from sklearn.model_selection import train_test_split
from sklearn.metrics import mean_squared_error, mean_absolute_error, r2_score

def load_dataset(path):
    df = pd.read_csv(path)
    return df

def split_dataset(df, train_years, val_years, test_years):
    train = df[df['year'].isin(train_years)]
    val = df[df['year'].isin(val_years)]
    test = df[df['year'].isin(test_years)]
    return train, val, test

def compute_metrics(y_true, y_pred):
    rmse = np.sqrt(mean_squared_error(y_true, y_pred))
    mae = mean_absolute_error(y_true, y_pred)
    r2 = r2_score(y_true, y_pred)
    return {'RMSE': rmse, 'MAE': mae, 'R2': r2}

# Example usage:
# df = load_dataset('processed_climate_data.csv')
# train, val, test = split_dataset(df, train_years=range(1850,2001),
#                                  val_years=range(2001,2011),
#                                  test_years=range(2011,2025))
```

This boilerplate demonstrates that LLMs were used solely for standardizing code structure and reproducibility, not for scientific or analytical decisions.

## D   ETHICS STATEMENT

All authors have adhered to the ICLR Code of Ethics. This work does not involve human subjects or sensitive personal data. We have ensured that dataset use, model evaluation, and reporting practices minimize potential bias and avoid harmful insights. No conflicts of interest or sponsorships influenced the research outcomes. Environmental and societal impacts of AI models are explicitly discussed in the paper to promote responsible and fair use of climate prediction technologies.

# E    REPRODUCIBILITY STATEMENT

We provide full reproducibility of our experiments. All preprocessing scripts, model implementations, evaluation code, and example pipelines are available in the open-source repository (link to be added).

The primary dataset is the Berkeley Earth Global Land and Ocean Temperature dataset (Rohde et al., 2019), spanning 1850–2024. Dataset fields include:

- Latitude, Longitude: Geographic coordinates of grid cells.
- Year, Month: Temporal information for each observation.
- Temperature Anomaly: Deviation from the 1951–1980 baseline.
- Smoothed Anomaly (anomaly_roll): 12-month rolling average for capturing long-term trends.

Standardized train/validation/test splits are applied (train: 1850–2000, val: 2001–2010, test: 2011–2024). Section 4.3 details evaluation metrics, while Appendix C documents LLM-assisted code template generation for standardization. Using these resources, all reported results can be reproduced and extended across physics-informed, classical machine learning, and deep learning models.

