# OpenReview forum: "From Physics-Informed Models to Deep Learning: Reproducible AI Frameworks for Climate Resilience and Policy Alignment"
_ICLR.cc/2026/Conference — ICLR 2026 Conference Desk Rejected Submission_

### Official Review · Reviewer_x5Aq · 2025-10-16

**Soundness:** 2
**Presentation:** 2
**Contribution:** 2
**Rating:** 2
**Confidence:** 3

**Summary:**

The paper proposes a reproducible benchmark for climate anomaly prediction. The authors standardize Berkeley Earth with fixed spatial–temporal aggregation and explicit train/validation/test splits, which is valuable given the inconsistent preprocessing that often undermines comparability across studies. They also provide a range of baselines that span physics-guided (LPS), classical ML (Random Forest, XGBoost), and modern deep learning (MLP, Transformers), enabling strong comparisons. Additional evaluation metrics are provided including standard RMSE/MAE, more predictive R², NSE, inference latency, energy use, and carbon footprint. This makes accuracy and efficiency trade-offs visible. Finally, the explicit quantification of energy and emissions and the framing against U.S. AI/climate policy goals grounds the work in real-world constraints. Overall, this contribution prioritizes transparency and practical relevance for future by formalizing emissions reporting alongside the stated metrics.

**Strengths:**

- The paper provides a strong direction towards the need of benchmark setups for easy in reproducibility of models and evaluation in climate domain.
- The use of additional evaluation metrics such as R2, NSE strengthens the quantification of errors in terms of anomaly
- The paper addresses important impact metrics as well such as training energy consumption, carbon emissions, training time, and inference latency which is essential in model selection and aids in trade off for accuracy and efficiency in model selection.
- The paper also addresses important aspects of climate modelling in terms of policy making.

**Weaknesses:**

- As a benchmark, the paper is missing the key details of the models, dataset, training hardware used in the experimental setup. Please refer to the questions section.
- In discussion section it is mentioned that  physics-informed models provide an optimal balance of accuracy, interpretability, and energy efficiency for regional temperature anomaly prediction, however only 1 physics based model is explored and in my opinion it is not enough to support this argument.

Minor typo:
Line 323 is missing a reference

**Questions:**

1. The authors mention the use of Berkley dataset due to its high resolution, however what is the resolution of the dataset used?
2. What is the input to the model (the shape or resolution of global data) and what is the output of the model (the resolution of the region)?
3. Which and how many hardware resources were used to train the models (this is also important to understand the emissions)?
4. The architectural details of all the models should be clearly mention to understand the performance metrics.
5. The benchmark setup should clearly state hyperparameters for each model's training such as (batch size, optimizers, schedulers etc).
6. For physics based models can you provide any experimental results on PINN models?

---

### Official Review · Reviewer_HFgL · 2025-10-28

**Soundness:** 2
**Presentation:** 1
**Contribution:** 1
**Rating:** 2
**Confidence:** 5

**Summary:**

This paper considers the Berkeley Earth dataset and benchmarks several methods for the task of regional anomaly detection including  a physics-informed model, machine learning and deep learning methods. They find that the physics-informed model outperforms the other methods and they also report the computational cost and energy consumption and carbon emissions of training the different methods. They also offer a reflection on implications for policy – that the focus on developing larger models might not be  necessarily be what is needed.

**Strengths:**

This is a project that considers a problem in a somewhat “holistic” way, and that benchmarks methods with considerations about environmental impact and broader downstream implications in terms of informing policy.
It can be appreciated that the authors benchmark simple ML models, deep learning models and a physics informed model and that they find that the simpler model outperforms the other methods, and that it seems like they have an approach which is very application-driven.

**Weaknesses:**

The main concern with this paper is the claims of contribution which are overstated. It sounds like the authors are presenting  benchmarking, evaluating models, including computational cost, and open sourcing code as a novel thing in itself, which is not.
It is actually a practice that has already been recognized as needed for reproducibility. The authors are therefore not p^resenting an AI reproducibility framework in itself, by benchmarking models on an existing dataset...
Moreover computing computational cost and emissiong  of training models is not new. There is no mention of existing work in that space (e.g. CodeCarbon package). In the way this paper is presented, it sounds like authors are trying to reinvent the wheel.

For example, the challenges outlined in 3.4  are not  addressed by the author’s proposal.
To only mention one thing, L.166 reads that one of the challenges is :
“Managing disparate datasets (Berkeley Earth, NOAA, EPA) with different formats”
However the authors address this by considering only the Berkeley Earth dataset. This is not really offering a solution for managing and aligning different datasets to choose to work only with one.
Another challenge that is mentioned is "“Mitigating geographic and temporal data biases”  which the authors do not do anything about.

Relatedly:
“While other datasets such as NOAA’s Climate Data Online (NOAA National Centers for Environmental Information, 2021) and EPA’s Envirofacts API (U.S. Environmental Protection Agency, 2020) offer complementary perspectives, we focus exclusively on Berkeley Earth to ensure reproducibility and consistent model evaluation.”
→ there is nothing that could prevent using multiple datasets and ensuring reproducibility

While it is appreciated that the authors make the code open-sourced, releasing dataset preparation code and experiments code is not a new thing in itself. Many works release code and dataset, especially in the context of benchmarking methods (cf. NeurIPS dataset and benchmarks track papers)

**Questions:**

In addition to the comments outlined in the weaknesses, there are other things I would like to highlight:

- Implementation details and training details are missing. Was there any HP tuning done for the deep learning method? What were the parameters chosen for the ML methods?

- What are the values chosen for estimating the carbon and energy emissions? I would encourage the authors to refer to existing works in the literature. Computation of emissions varies based on GPU type, country, power grid, … The authors do not even specify what kind of GPU is used.

Minor comments:
- The figures are not necessarily informative, and the text description could suffice. Also, the figures are taken from existing sources.
- L323: missing reference
- The discussion about policy is not very grounded. It is fine to include it as part of the conclusion, but making it a whole section in itself, one would expect the work to be placed better in the broader context of reflections on this topic. that last part shares an interesting position but while I agree with this position, it seems like this should be a paper of its own, that cannot rely on just a single example on one dataset.

I do appreciate the considerations that the authors have about the environmental and policy impacts of this work but in the current state, the paper seems to be exaggerating the contribution it has. There is insufficient literature review and relation to existing work.

---

### Official Review · Reviewer_GhvF · 2025-10-30

**Soundness:** 2
**Presentation:** 3
**Contribution:** 1
**Rating:** 2
**Confidence:** 4

**Summary:**

The paper introduces a framework for assessing the ability of simple climate models to predict temperature anomalies. In addition to accuracy, other metrics, such as energy usage and CO2 emissions, are implemented. Overall, the framework seems to work and suggests that physics-based neural networks are best performing for the setup.

**Strengths:**

The contribution is timely, and there's undoubtedly a need for good metrics and frameworks that help assess ML-based climate models with respect to more than just accuracy. The presented framework is certainly going into that direction.

**Weaknesses:**

In an application scenario, such as the one described related to policy making, one would also have the possibility to either downscale the non-local climate information or use pre-trained climate models, either out-of-the-box or with some fine-tuning. The scenario, where one would from scratch train their own climate model, seems a bit far-fetched at this point. All the analysis related to energy consumption, etc., would then significantly change.
Additionally, I am a bit confused about the data setup. The anomalies plotted in Figure 3 look like yearly anomalies, making this an incredibly small dataset especially for the transformer model. The comparison thus doesn't really seem useful (as the model is just not build for this kind of small data set).

**Questions:**

- What's the training size?
- Who would train their own climate model for temperature anomalies only?
- All the metrics are heavily dependent on the compute setup. How would this generalize across setups?

---

### Official Review · Reviewer_C7oT · 2025-11-03

**Soundness:** 1
**Presentation:** 1
**Contribution:** 1
**Rating:** 0
**Confidence:** 5

**Summary:**

This paper aims to provide a benchmark for prediction of regional temperature anomalies from global temperature anomalies. It introduces a train/test split and feature/label creation strategy for the Berkeley Earth dataset, and runs several baselines on the data (linear pattern scaling, random forest, XGBoost, MLP, and transformer). Metrics compared capture a combination of predictive accuracy and computational efficiency/carbon footprint. The simpler linear pattern scaling model is found to perform best across both sets of metrics.

**Strengths:**

* The motivation of providing a benchmark for an important domain (climate prediction), and of measuring both performance and computational efficiency, is valid.
* The motivation of identifying whether simpler models outperform more complex models is valid, rather than assuming more complex models will necessarily work better.

**Weaknesses:**

* The task definition does not capture a task that is necessarily well-suited for machine learning, given the use of a one-dimensional input and a one-dimensional output. Machine learning is generally best suited for finding nuanced correlations between multiple features, rather than learning a scalar-to-scalar mapping.
* The task definition is rather limited in scope, capturing only one output variable (temperature anomaly) among many that could be captured for this dataset. The reason for this limitation is not clear.
* The dataset used is an established dataset (Berkeley Earth), and the proposed benchmark does not provide sufficient additionally in terms of combining multiple datasets or otherwise presenting the data or task in a meaningfully novel way.
* While one of the contributions posed is an automated data preprocessing pipeline, sufficient detail is not provided to evaluate this contribution. In particular, the paper lacks sufficient depth in describing the novelty of this pipeline, and code is not provided.
* The abstract promises an in-depth policy analysis, but instead, only a few sentences of high-level commentary are provided towards the end of the paper.

**Questions:**

What is the motivation for limiting to a scalar-to-scalar mapping, focusing on temperature anomaly only, and leveraging the Berkeley Earth dataset only? (While some motivations such as "to ensure reproducibility and consistent model evaluation" are given, these motivations do not make sense, as reproducibility and consistent model evaluation are possible in more involved settings as well.)

---

### Note · Program_Chairs · 2026-01-17
**Submission Desk Rejected by Program Chairs**

The following references in this submission do not refer to real documents and/or have major errors in bibliographic information:

 A. Williams and R. Gupta. Ai infrastructure and climate ethics. AI and Society, 36(3):421-432, 2021.
B. Ruddell and C. Monteleoni. Overfitting in climate-ai models: Challenges and risks. Climatic Informatics, 2020.
Thomas Hauser, Corina Sandu, Brian Soden, et al. Artificial intelligence in climate science: Opportunities, challenges, and future prospects. Nature Climate Change, 12(5):371-381, 2022.
C. Monteleoni and B. Ruddell. Deep learning applications in weather and climate science. In Proceedings of AAAI, 2018.
J. Smith and T. Brown. Overfitting in climate prediction models: Risks and mitigation. In Proceedings of NeurIPS Workshop on Climate Informatics, 2021.
C. Huntingford, J. A. Lowe, B. B. Booth, and C. D. Jones. Contributions of climate model uncertainties to future projections of lps (linear pattern scaling). Climatic Change, 2012.